# Effects of Embarrassment on Self-Serving Bias and Behavioral Response in the Context of Service Failure

**DOI:** 10.3390/bs14020136

**Published:** 2024-02-14

**Authors:** Kai-Chieh Hu, Hsin-Lin Tsai

**Affiliations:** Department of Business Administration, Soochow University, Taipei 10048, Taiwan

**Keywords:** embarrassment, self-serving bias, service failures, behavioral response

## Abstract

Previous research has focused on examining embarrassment in sensitive product purchase situations. Although embarrassment is a widespread emotion in consumption situations, few studies have explored its impact on service encounters, especially in the service failure context. This study examines how customers react to different service failures that cause embarrassment and explores whether self-serving bias exists when customers perceive higher embarrassment in service failure. This study uses a 2 (source of failure) × 2 (level of embarrassment) scenario experimental method to examine the effect of two sources of failure on consumer locus attributions, negative emotions, and negative behaviors, considering the moderating effects of the level of embarrassment. Data were collected from 218 student subjects in Taiwan. The results show that embarrassment is important in service failure contexts. Specifically, when consumers perceive higher embarrassment, they attribute more responsibility to the service provider. These attributions, in turn, influence customers’ emotions and behavioral responses. These findings have several important theoretical and practical implications in terms of embarrassing service failures.

## 1. Introduction

Services are generally inseparable, which means that they are produced and consumed simultaneously [1]. Contact points where a customer contacts the service producer are frequently called “moments of truth” or service encounters [2]. The heart of a service is the interaction between the server and customer [3]. This is where emotions meet economics in real time, and where most customers judge the service quality. Service encounters involve numerous uncontrollable elements. Service failures, thus, are commonplace and considered an inevitable consequence of service provision [4,5]. When service failures occur, customers will seek explanations, and these attributions affect their emotional responses [6]. The consequences of emotional responses may lead to negative word-of-mouth, dissatisfaction, and defection [7,8].

Customers facing service failure situations may sometimes feel embarrassed and may even wish that the ground would open up and swallow them, or that time has stood still. Feelings of humiliation and discomfort are also aroused, albeit often only momentarily. Embarrassment is thus a sufficiently unpleasant emotion that people will go to great lengths to avoid associated actions or situations [9]. Numerous reasons explain embarrassment in this situation, including a fear of facing criticism, looking like a fool, losing face, and so on [10]. Embarrassment can influence numerous facets of human social behavior and occurs across various consumer behavior contexts [11].

Since embarrassment is a negative and self-conscious emotion, it results from deficiencies in the presented self and implies a feeling of foolishness and awkwardness [12]. Service failure incidents make people highly sensitive to how others perceive them and desirous to avoid embarrassment [13,14]. Therefore, people may make self-serving attributions in order to present a certain image of themselves to others [15]. Customers are thus more likely to blame service employees to protect their self-esteem, even for failures that are caused by themselves. This behavior is called “self-serving bias”. Self-serving bias is defined as the tendency to attribute success to the self and failure to external factors [14,16]. Self-serving bias has been demonstrated in decision making contexts [17]. Service failures thus make consumers feel embarrassed, which may lead them to blame failures on someone or something else to avoid causing public embarrassment [14,18].

Specifically, embarrassment is also a consumption emotion that is highly relevant to face-to-face service encounters [10] and is a strong motivating force affecting decisions [19]. Previous studies have focused on examining embarrassment in relation to the purchase of sensitive products [11,20,21] or coupon usage [22,23]. However, there has been a gap in research, as there has been no prior investigation into the role that is played by embarrassment in service failures and its impact on variables such as service failures and locus attribution. This study thus aims to examine the role of embarrassment in service failure and explores how different sources of service failure influence consumer embarrassment and attribution tendency, as well as the consumer emotional and future behavioral response.

## 2. Literature Review

### 2.1. Service Failure

Services are produced and consumed simultaneously, which means that service provision occurs in public, and the interaction between the service provider and customer is visible. Owing to the significant interpersonal contact, service failures are inevitable [5]. Service failure is commonly defined as a mistake, problem, or error during the process of service delivery [4]. In this situation, service performances that fall below customer expectations cause customer dissatisfaction during service encounters [24].

Service failure can thus produce strong negative emotions in customers that may negatively influence customer satisfaction and future behavioral intentions [25]. Research findings have confirmed that service failures can cause customer dissatisfaction, and in turn provoke customer anger, complaint intentions, negative word-of-mouth, and defection [6,7,8,26].

Service failures may vary in severity, frequency, and timing [26], and hence, their classification is essential. Divergent classifications of service failure categories exist. Bitner et al. [4], focusing on service encounters, employed a critical incidents technique (CIT) to identify the source of customer dissatisfaction in various service settings (such as hotels, restaurants, and airline industries). Their study classified three main groups of perceived service failures: employee response to service delivery system failures, employee response to customer needs and requests, and unprompted and unsolicited employee actions.

Furthermore, Bitner et al. [27] identified service failures from the perspective of employees and added a new group of service failures called Problematic Customer Behaviors. Their research indicates that unsatisfactory service encounters may result from inappropriate customer behaviors. Subsequent researchers have adopted these categories of service failures as the basis of classification and have extended them to other service failure issues [28,29,30]. Many studies have argued that service failure can occur either in the core services (outcome failures) or during service delivery processes (process failures) [8,28,31].

Because of the nature of service, customers and relevant service components (including the service provider, service process, and servicescape) come together to facilitate simultaneous service production and consumption. Therefore, the above studies indicate that service failure can result not only from service providers but also from customers themselves. This study broadly classifies the source of failure into service provider failures and customer failures.

### 2.2. Service Failure and Attributions Theory

Attribution theory states that people are rational information processors whose causal inferences influence their actions [6]. This shows how people explain their experiences, the information that they used in making causal inferences, and how these interpretations influence subsequent evaluations and actions [32,33]. Attribution theory is generally seen as originating from the work of Heider [34]. The existing research argues that causes can be attributed to factors within the person (internal factors) or the environment (external factors). Weiner et al. [33] redefined this internal–external distinction as the locus of the attribution dimension. The locus of attribution states that if something goes wrong in a service encounter, the customer will try to assign responsibility. When the consumer accepts the blame for the service failure personally, this is termed internal attribution, while if the consumer blames the service provider, this is external attribution [35].

In response to service failures, consumers frequently attempt to determine their causes and make attributions based on received or perceived information [36,37,38]. It is generally accepted that when a service provider causes service failure, the consumer will attribute more responsibility to the service provider than to themselves. This study thus proposes the following hypothesis:

**H1.** 
*Different sources of service failure influence consumer locus attribution. When a service provider causes service failure, the consumer is more likely to attribute responsibility to the service provider than to themselves.*


### 2.3. Service Failures and Embarrassment

Embarrassment is sufficiently uncomfortable that people will generally go to great lengths to avoid it [9]. Embarrassment can be defined as a commonly occurring, short-lived, negative emotional response to a threat to the public self in front of a real or imagined audience [39,40]. Embarrassment seems to result from a human concern with what others may think [12]. Thus, embarrassment differs from other self-conscious emotions (such as shame and guilt), because incidents provoking embarrassment nearly always occur in public (i.e., one rarely feels embarrassed alone) [41]. Moreover, embarrassment threatens the perceived social identity of an individual.

As a negative self-conscious emotion, embarrassment was found to cause negative self-evaluation or unwanted self-exposure, which is associated with a loss of public self-esteem. Once embarrassment occurs, people feel awkward, flustered, or mortified, which further stimulates their intention to flee the situation [42]. Additionally, embarrassment is an unpleasant experience that may occur in various situations, including consumption situations [11]. For example, embarrassment may occur during product purchase (e.g., buying adult videos, condoms, and other unmentionables), in usage situations (e.g., when one’s credit card is denied while paying the bill at a high-end restaurant), or during service consumption (e.g., physical check-ups or weight loss services) [43].

The heart of a service is the interaction between the server and customer [3]. Therefore, service providers, customers, and others cooperate to facilitate service production and consumption. Owing to the human elements, service failures are common and considered an inevitable consequence of service provision [4,5,44]. Embarrassment is a face-threatening element that can arise from service failures. In a service failure situation, customers may feel a threat to their self-esteem owing to embarrassment and the consequent loss of face, particularly when other consumers witness the service failure [43]. Grace [10] conducted an exploratory study to gather data regarding embarrassing consumption situations and yielded the functional analysis of consumer embarrassment (FACE) model. This model identified three basic sources of embarrassment in service interactions: the service provider, the focal customer, and fellow customers [22,45,46]. Thus, an embarrassing failure occurs when a consumer perceives that a service is not delivered properly in a consumption context and develops an aversive and awkward emotional state that increases the threat of unwanted evaluation from real or imagined audiences. Furthermore, this model examined consumer reactions to embarrassment, including emotional, physiological, and behavioral reactions [22].

Service failure can be attributed to external factors (service providers) or internal factors (customers themselves). However, previous findings demonstrate that consumers may also make mistakes that are likely to cause embarrassment during service delivery. Customers may thus generate feelings of foolishness and awkwardness [12,46]. In this situation, customers are more likely to exhibit self-serving bias to protect their self-image [15]. Customers thus blame service providers to avoid losing face. Despite embarrassment being common in consumption, few studies have examined its influence in service failure contexts. This study thus focuses on the embarrassment that may result from service failure. The source of failure can be classified broadly into service provider-induced failures and customer-induced failures.

Embarrassment is a consumption emotion that is particularly relevant to face-to-face service encounters [10]. This emotion is important in consumer behavior, which is a strong motivator affecting numerous decisions and judgments [11,19]. Customer embarrassment may even induce negative word-of-mouth intentions. This dark side of embarrassing situations can cause misunderstandings and lost opportunities for firms [21,23,41]. This study thus attempts to explore the attribution of locus in relation to customer embarrassment in service encounters.

If a service failure is the fault of the customer, they will personally accept blame. Grace [22] showed that emotions can derive from different sources. The present study examines whether the level of consumer embarrassment depends on the sources of failure. For example, when consumer actions cause a service failure, the consumer themselves may experience higher embarrassment. This may occur because customers care more about saving face and are concerned about how others perceive them. Customers may thus internalize their embarrassment more than if the source of failure is external. This study thus proposes the following hypothesis:

**H2.** 
*Different sources of service failure influence customer embarrassment. When a service provider causes service failure, the embarrassment to the consumer may be less than when the embarrassment is over their own action.*


### 2.4. The Moderating Effect of Level of Embarrassment between Service Failure and Locus Attribution

However, given limited information regarding the real cause of a service failure, it has been shown that customers frequently blame failures on service providers or other factors, even when they are the fault of the customer themselves [18]. The attribution literature has demonstrated that people tend to exhibit self-serving bias. Self-serving bias here describes the tendency of individuals to attribute positive events to themselves, while attributing negative events to other causes [47]. That is, humans tend to take credit for successes but deny responsibility for failures. This bias is motivated by the desire to maintain a positive self-image and results in selective interpretation of reality.

Self-serving bias has been identified in numerous contexts. For example, this self-serving bias has been demonstrated in achievement situations [16]. People exhibit self-serving bias regarding their workplace performances. The self-serving bias has also been applied to the service failure situation by Bitner et al. [27], who showed that it led customers to blame service providers or the system, whereas service providers tended to blame the system or customers. The result was employees and customers having different views regarding the cause of service dissatisfaction.

Several reasons have been proposed to explain the occurrence of self-serving bias. Some researchers have proposed a motivational explanation [16]. One explanation of this perspective is that self-serving biases reflect the need to engage in self-enhancement [14]. The benefits to self-worth motivate individuals to protect their self-esteem. Individuals taking personal responsibility for undesired outcomes reduce their self-worth [15]. People are motivated to favorably impress others, and thus, they take personal responsibility for successes but not failures to influence how others perceive them [48]. Based on the self-serving bias perspective, individuals tend to alter their perceptions of causality to protect their self-esteem. Therefore, customers encountering service failures tend to blame failure on the service provider, even when it is their own fault.

When service failures occur, most customers will look for who is responsible for the failure. However, it has been shown that customers often blame failures on service providers or something else, even if the failure is due to the customer’s fault [18]. The attribution literature shows that people tend to engage in “self-serving bias”. This bias has also been established in consumer behavior contexts. When faced with a negative event, consumers will tend to form biased attributions in order to protect their self-esteem [49].

Service failure situations cause customers to feel a threat to their social identity that causes embarrassment. Previous studies have proposed that embarrassment is a sufficiently discomforting emotion. Embarrassment makes people feel awkward, flustered, or mortified and may inspire a desire to flee the situation [42]. Consumers thus may be more likely to exhibit self-serving bias to avoid embarrassment and protect their self-esteem. Thus, the influence of sources of failure on locus attribution could be moderated by the level of embarrassment. When consumers experience higher embarrassment in service failure, they are more likely to attribute responsibility to service providers than to themselves, even when the failure is their own fault. This study thus proposes the following hypothesis:

**H3.** 
*The level of embarrassment that is felt by customers moderates the relationship between the sources of service failure and locus attribution. When a service failure is caused by a consumer themselves, the consumer is more likely to attribute responsibility to the service provider when they perceive high embarrassment than when they perceive low embarrassment.*


### 2.5. The Influence of Locus Attribution and Level of Embarrassment

Previous studies proposed defining embarrassment as a negative emotional response that occurs when an individual feels threatened by another person [19,23,50]. People thus go to great lengths to avoid this situation. Feelings that are possibly related to embarrassment include humiliation, awkwardness, fluster, and mortification. Embarrassment is thus a discomforting experience that may be associated with negative feelings. Consumers with high embarrassment generate stronger negative emotions. This study thus proposes hypothesis four as follows:

**H4.** 
*A higher level of embarrassment positively influences negative emotion.*


Previous research has clearly demonstrated that the locus of attributions affects several important affective and behavioral outcomes [4,6]. Numerous studies have found that external attribution causes several negative consequences, including consumer anger, negative word-of-mouth, and consumer feelings of deserving a refund and an apology [6]. On the other hand, internal attribution generates guilt and regret that tend to result in unsatisfied customers doing nothing [36]. This study thus further explores the effect of the locus of attribution in embarrassing service failure. This study proposes that the consumer’s locus of attribution influences the customer’s negative emotion and behavior. In this study, negative emotion refers to unfavorable emotional experiences towards the service provider. When service failures stem from the actions of the service provider, consumers may feel anger, disappointment, or frustration. It could also increase the likelihood of complaining, changing intentions, and spreading of negative word-of-mouth. This study thus hypothesizes the following:

**H5.** 
*External locus attribution positively influences negative emotions.*


**H6.** 
*External locus attribution positively influences negative behaviors.*


Service failures frequently evoke strong customer responses. Previous studies have shown that consumers’ affective response to service failures influences their service evaluations [6]. According to Grace [10], embarrassment leads to negative word-of-mouth and negatively impacts future patronage. Therefore, in response to an embarrassing service failure, consumers perceive stronger negative emotions, and these experiences are expected to significantly and negatively impact future patronage. Consumers thus become more likely to complain, have a stronger intention to switch, and are more likely to spread negative word-of-mouth. This study thus proposes one final hypothesis as follows:

**H7.** 
*Negative emotion positively influences negative behaviors.*


Drawing upon the aforementioned literature review, this article introduces a conceptual framework, which is illustrated in Figure 1. This study examines how the various sources of service failure influence customers’ response in the context of consumer embarrassment. In this framework, the source of failure can be classified broadly into the service provider and consumer. Different sources of service failure influence consumers’ locus attribution. On the other hand, this study argued that an embarrassing service failure can contain self-serving bias. The influence of self-serving bias on locus attribution depends on the level of embarrassment that is involved in the service failure. Additionally, the locus attribution influences negative emotions and behaviors.

## 3. Method

### 3.1. Experimental Design

This paper used a scenario experimental design for several reasons. First, this method permits the inclusion of a representative set of service failure situations. Second, it minimizes memory bias, which is common in self-reports of service failures in survey designs. Finally, this method enhances internal validity by allowing researchers to control extraneous factors that may influence the study results [31,51].

A 2 (source of failure: service provider and consumer) × 2 (level of embarrassment: high and low) factorial between-subjects design was employed to test the study’s predictions. Restaurant scenarios were appropriate, because service failures are common in this industry and embarrassment easily occurs during interpersonal customer-to-employee encounters in a restaurant.

Restaurant scenarios were thus used to manipulate the level of embarrassment and source of service failure. The scenarios involved asking participants to imagine that they had reserved a nice window table at a high-end restaurant to celebrate the birthday of their girlfriend (or boyfriend) or for a get-together with old friends. In the service provider failure condition, the service provider informed the subject that the restaurant had recorded the wrong reservation date, and thus, there were no tables available and the participant would have to wait. In the consumer failure condition, the subject imagines that they have arrived at the restaurant 20 min late and are told by the waiter that their reservation has been cancelled and that no window view tables are available.

Additionally, based on prior studies, this study suggests that the presence of other consumers is a sufficient condition that creates high embarrassment in service encounters [11,43]. Thus, in the high-embarrassment situation, subjects were told that they planned to celebrate the birthday of their girlfriend (or boyfriend) at the restaurant and saw other consumers witness the service failure. Meanwhile, in the low-embarrassment situation, subjects were enjoying a gathering with friends and did not perceive other consumers to have witnessed the service failure.

Participants were randomly assigned to either a service provider- or customer-induced embarrassing failure condition. At the beginning of the experimental session, participants were asked to read a scenario describing an incident of service failure in a restaurant. After reading the scenario, participants were asked to respond to a series of questions about their level of embarrassment, attributions of the cause of the failure, negative emotions, and negative behavioral response.

### 3.2. Measures

All the manipulation checks and dependent measures were measured using a 5-point Likert scale (1 = strongly disagree; 5 = strongly agree). Level of embarrassment was measured using three items, drawn from previous research [9,11,12]. The items were anchored using the following labels: embarrassed, uncomfortable, and awkward (such as, “You feel very embarrassed about the incident”). Locus attribution measures the degree to which participants attributed the source of service failure to two different parties: the service provider (“the service providers should take responsibility”, “the service providers caused the outcome”, “I would blame the service providers for the outcome”) and consumers themselves (“the outcome was my fault”, “I should take responsibility”, “I would blame myself”). Negative emotion was measured using three items, adopted from Grace’s FACE Model [10]. Anger, humiliation, and unhappiness were the most commonly reported feelings during the embarrassing incident (60%). This study measured negative behavior using three items, namely, intention to complain, switch, and spread negative word-of-mouth [43]: “I will complain to my family and friends about the incident”, “I will tell my family and friends not to go to the restaurant”, and “I will never visit the restaurant again—at least in the near future”.

### 3.3. Data Collection

Data were primarily collected from university students in Taiwan. The sample comprised 218 respondents, of whom 58% were female. Because employing students as major participants is a common practice in experimental studies, and since students possess homogeneous backgrounds, numerous extraneous variables such as age, education, and occupation can be controlled. The use of student participants thus helps remove potential bias in non-student samples and achieve internal validity. Furthermore, students are real-life consumers of restaurant services. In this study, participants were randomly assigned to one of four scenarios. All participants were told that this was a study on customer service behaviors and were asked to complete the questionnaires carefully.

Reliability analysis was used to examine the internal consistency of different variables under one dimension. Reliability analysis was tested using Cronbach’s coefficient alpha. Generally, a Cronbach’s alpha coefficient exceeding 0.7 can be considered high reliability [52]. Table 1 shows that the Cronbach’s α of each item exceeded the criteria of 0.7, indicating that the survey had satisfactory internal consistency reliability.

## 4. Results

### 4.1. Manipulation Checks

To ensure that the manipulation was effective, this study compared the means for the three-item level of embarrassment scale. The results from an independent *t*-test indicate a significant difference in ratings (*t* = 12.392, df = 216, *p* < 0.05) between the high-embarrassment (M = 4.23) and low-embarrassment conditions (M = 2.77). Hence, the analytical results indicate that the manipulation was successful.

### 4.2. The Effect of Service Failure Sources on Locus Attribution

Hypothesis 1 suggests that when the service provider causes the service failure, the consumer is more likely to attribute responsibility to the service provider than to themselves. To test this hypothesis, this study conducted independent sample *t*-tests to examine whether different sources of service failure influence the consumer’s locus attribution. The results indicated that subjects in the scenario in which the service provider was to blame attributed more responsibility for the service failure to the service provider than they did those in the consumer failure scenario (*t* = 11.96, *p* < 0.05). On the other hand, subjects in the consumer failure scenario felt themselves to be more responsible for the service failure than those in the service provider failure scenario did (*t* = 17.448, *p* < 0.05). Therefore, subjects in the two sources of service failure differed significantly in their locus attribution to the service provider or to themselves. Hence, Hypothesis 1 is supported.

### 4.3. The Effect of Service Failure Sources on the Level of Embarrassment

Hypothesis 2 addresses the effect of service failure sources on the level of consumer embarrassment. An independent sample t-test examined whether the level of consumer embarrassment differs with the source of service failure. The analysis revealed that the degree of embarrassment that subjects experience differed significantly across the two sources of failure (*t* = 2.607, *p* = 0.01 < 0.05). The results clearly show that consumers are more likely to experience more embarrassment when the service failure results from their actions than when it is caused by the service provider. Hence, Hypothesis 2 is supported.

### 4.4. The Moderating Role of the Level of Embarrassment

Hypothesis 3 proposed that the effect of the sources of failure on locus attributions is moderated by the consumer’s level of embarrassment. Multiple regression analysis was performed to test the moderating effects in this section. Table 2 lists the results of the multiple regression analysis. Model 3 states that level of embarrassment moderated the relationship between the sources of failure and the attribution to the service provider. The data supported this interaction (B = 0.315, *t* = 2.931, *p* = 0.004).

Figure 2 details this significant interaction. The graph shows that when consumers perceive higher embarrassment, associated with service failure, they will attribute more responsibility to the service provider, an effect that is stronger in the condition of consumer failure. Specifically, the results show that this reinforcement effect is stronger for the condition of consumer failure and shows that the scores for attribution to the service provider are 1.7 for consumers in the low-embarrassment condition versus 2.7 for those in the high-embarrassment condition. Therefore, the level of embarrassment appears to strengthen the effect of attribution to the service provider; when service failures occur, the consumer is more likely to blame them on service providers when they perceive high embarrassment. This moderating effect of the level of embarrassment is stronger for consumer failures than for service provider failures. Hence, Hypothesis 3 is supported.

### 4.5. The Relationships between Locus Attribution, Level of Embarrassment, Negative Emotions, and Behavioral Response

This study used structural equation modeling (SEM) to assess the appropriateness of the framework and examine the relationships between locus attribution, level of embarrassment, negative emotion, and negative behavior. The first step applied confirmatory factor analysis (CFA) to refine a set of scales to represent the model constructs. The second step then applied path analysis to determine the relationship among variables that were included in the structural model.

To evaluate the goodness of fit in CFA, it is suggested that the chi-square normalized by degrees of freedom ($\chi^{2}/df$) should not exceed 5, and that GFI, AGFI, NFI, IFI, NNFI, and CFI should exceed 0.9. Furthermore, SRMSR, RMR, and RMSEA should be less than 0.05 [53]. The initial results of the CFA in this study show that $\chi^{2}/df$ = 3.044 (146.094/48), GFI = 0.91, AGFI = 0.85, NFI = 0.92, NNFI = 0.93, IFI = 0.95, CFI = 0.95, RMSEA = 0.097, and SRMR = 0.079. The modification index (MI) indicates that V3 is closely correlated with other variables. That means that V3 is a complex variable (measuring multiple latent variables) and should be removed from the measurement model [54]. After eliminating V3, the values of fitness indices become better than the original value. $\chi^{2}/df$ = 2.63 (99.878/38), GFI = 0.93, AGFI = 0.88, NFI = 0.94, NNFI =0.94, IFI = 0.96, and CFI = 0.96. Additionally, RMSEA = 0.087, and SRMR = 0.065. These results indicate an acceptable goodness of fit for this model [55].

Moreover, Table 3 lists the reliability and validity results of the measurement model. All standardized factor loadings exceed 0.5, and each indicator t-value exceeds 1.96, indicating convergent validity. Additionally, the average variance extracted and composite reliability were estimated for each scale. Each AVE exceeded the minimum value of 0.5 suggested by Fornell and Larcker [56]. To examine individual construct reliability, the composite reliability for each construct exceeded the recommended acceptable level of 0.7. Thus, the model’s reliability was confirmed.

Table 4 lists the results of the hypothetical relationships. The results of the path analysis support the four hypotheses (H4–H7). The results indicate that embarrassment is experienced as a feeling of discomfort. Consumers with high embarrassment thus generate a stronger negative emotion, and so, Hypothesis 4 is supported (*β* = 0.36, *t* = 5.46, *p* < 0.001). The results also demonstrate statistical support for Hypotheses 5 (*β* = 0.51, *t* = 7.39, *p* < 0.001) and 6 (*β* = 0.14, *t* = 2.07, *p* < 0.05), suggesting that external locus attribution positively and significantly influences negative emotions and behavioral responses. As posited, when a service failure occurs, consumers attribute responsibility to the service provider, which generates more negative emotions, such as anger, humiliation, and unhappiness and increases the likelihood of customer complaints, switching, and negative word-of-mouth. Finally, Hypothesis 7 is also supported (*β* = 0.79, *t* = 8.78, *p* < 0.001). The results reveal that negative emotion positively influences negative behavior. Consumers who respond to service failures with more negative emotion are thus more likely to complain, switch, and spread negative word-of-mouth.

## 5. Discussions

### 5.1. Managerial Implications

The findings of this study have several implications for management. This study underscores the significance of embarrassment in the context of service failures. The findings reveal that both service provider-induced and consumer-provoked embarrassment trigger physiological reactions, resulting in consumers feeling awkward, flustered, or mortified. Specifically, consumers may experience heightened embarrassment when their actions lead to service failure, and in situations with a high embarrassment level, consumers tend to display more negative emotions and react more adversely.

The practical management implications are substantial. Service providers must place considerable emphasis on understanding and addressing customer perceptions of embarrassment during service encounters, given its potent influence on negative behavioral responses. As service providers gain deeper insights into the impact of embarrassment on customer reactions to service failure, they can enhance their ability to navigate such situations effectively. For instance, managers should prioritize the recruitment of employees with strong communication and problem-solving skills. Additionally, targeted front-line employee communication skills training can empower staff to comprehend the effects of embarrassment on customers, enabling service providers to handle customers with sensitivity and professionalism, thereby minimizing embarrassment in service encounters. These proactive measures contribute to fostering a positive customer experience and mitigating potential negative outcomes associated with service failures.

Moreover, this study offers crucial insights into the influence of embarrassment on consumer attributions to service failures. Specifically, it highlights that in situations where service failures lead to embarrassment, the locus of attributions tends to demonstrate a self-serving bias. Consumers, in an effort to safeguard their self-image, often attribute more responsibility to service providers. This tendency is pronounced when a service failure significantly threatens their desired social identity, prompting them to adopt avoidance strategies.

In practical terms, when service providers contribute to embarrassing failures, a recommended approach is to employ effective service recovery strategies that restore consumers’ dignity. For instance, service providers can consider implementing public recovery measures, allowing other customers to witness the efforts that are taken. This method helps consumers in restoring their self-esteem by demonstrating the provider’s commitment to rectifying the situation.

Conversely, when service failures are initiated by consumers, it is imperative for service providers to refrain from ridiculing customers and ensuring they do not feel embarrassed for their mistakes. Instead, treating customers with sincerity and empathy is crucial. Service providers should actively work to soothe the level of consumer embarrassment and express a genuine willingness to assist in problem resolution. By transforming negative experiences into positive ones, service providers can effectively mitigate the adverse impacts of embarrassing service failures, thereby enhancing the likelihood of customer satisfaction. These practical strategies underscore the importance of empathy, sincerity, and strategic service recovery efforts in managing and improving customer experiences in the face of embarrassing service failures.

### 5.2. Suggestions for Future Research

This study, like all others, suffers various limitations that constrain generalization based on the findings and open up directions for future research. Firstly, the utilization of a scenario-based experiment, while maximizing internal validity, may hinder respondents from fully immersing themselves in the depicted situation. To overcome this limitation, future studies are encouraged to employ field survey approaches targeting real-life service failure scenarios.

Secondly, the study focused exclusively on a single service context (specifically, restaurants). Future research endeavors should diversify by exploring various service industries to assess the universality of embarrassment in customers’ responses to service failures. Also, the findings’ generalizability is confined to two types of embarrassing service failures—those caused by the service provider or the consumer. However, many service failures stem from the misbehavior of other customers. Future studies could adopt a more realistic perspective by incorporating third party-induced embarrassing failures.

Thirdly, future research should delve into individual differences in emotional responses, particularly regarding embarrassment. For instance, Honea [57] discovered that individuals with high public self-consciousness (PUBSC) exhibit a heightened inclination to avoid embarrassment, often altering their behavior to protect and enhance their positive self-image. Individuals with high public self-consciousness may be more susceptible to self-serving bias during embarrassing service failures. Consequently, future studies should investigate the role of public self-consciousness and its impact on consumers’ locus attribution in the context of embarrassing service failure. Examining these individual differences can contribute valuable insights into the nuances of emotional responses and attribution processes in the face of service failures.

Furthermore, due to the absence of prior research exploring the impact of embarrassment on service failures and attributions, this study primarily aims to validate and confirm the disruptive effects of embarrassment within the context of service failures. The study has not investigated other variables that may moderate the effect of embarrassment in this context. However, there may be other factors that could moderate or transform the impact of embarrassment on negative emotions or behaviors. Future research could explore the potential influence of other moderating variables, such as personality traits, based on the behavioral approach system (bas) [58], severity of errors, and so forth.

Finally, Wan [43] found that those with a collective mentality react more negatively to embarrassing service failure than individualists. It would be interesting to conduct a cross-cultural study to further examine consumers’ attribution for service failures involving embarrassment.

## 6. Conclusions

Based on the results of previous research, this study examined the role of embarrassment in service failure. This study also found that the locus of attributions reflects a self-serving bias that is dependent on the degree of embarrassment in service failures. First, this study showed that the sources of service failure significantly affect the level of embarrassment. The results indicate that the level of consumer embarrassment depends on who caused the service failure. Consumers experience less embarrassment when that embarrassment is triggered by the service provider than when it is triggered by themselves. Second, this study found that the level of embarrassment moderates the effects of the sources of failure on customers’ locus attributions. The research results demonstrate that when the consumer perceived a higher level of embarrassment associated with service failure, they attributed more responsibility to the service provider, despite the failure being caused by themselves. This indicates that whether the locus of attributions reflects a self-serving bias depends on the level of embarrassment that is involved. Thus, when service failures occur, a customer’s likelihood of exhibiting self-serving attribution bias increases with the level of customer embarrassment. Third, this study found that consumers with high embarrassment associated with service failure generate stronger negative emotions, including anger, humiliation, and unhappiness. Finally, the results show that the locus attribution significantly influences negative emotions and negative behavior. In a service failure, customers might feel a threat to their self-esteem that causes embarrassment. Therefore, they become more likely to attribute responsibility to service providers. This generates more negative emotions and increases the likelihood of complaining, switching intention, and negative word-of-mouth. Additionally, this study also found that consumers perceive stronger negative emotions that significantly and negatively influence future patronage.

## Figures and Tables

**Figure 1 behavsci-14-00136-f001:**
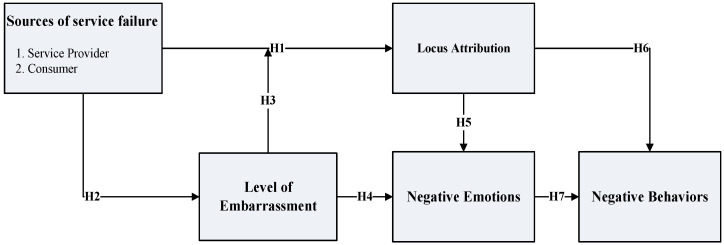
Research model.

**Figure 2 behavsci-14-00136-f002:**
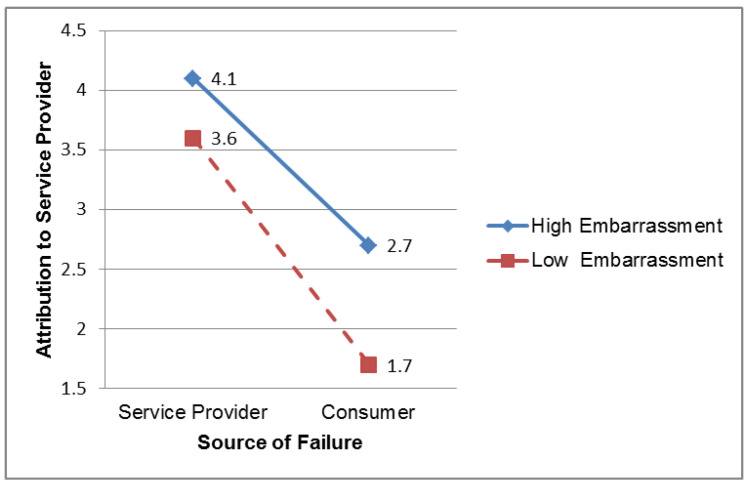
Moderating effect of embarrassment level on the relationship of failure source and attribution to service provider.

**Table 1 behavsci-14-00136-t001:** Reliability analysis.

Construct	Cronbach’s α
Attribute to service provider	0.905
Attribute to customer	0.931
Level of embarrassment	0.918
Negative emotions	0.716
Negative behaviors	0.861

**Table 2 behavsci-14-00136-t002:** Regression results for attribution.

Model	Variables	Beta	*t*-Value
1	(Constant)	3.877	43.172 *
	Source of Failure	−1.553	−11.940 *
2	(Constant)	2.789	14.266 *
	Source of Failure	−1.685	−13.800 *
	Level of Embarrassment	0.330	6.147 *
3	(Constant)	3.210	13.383 *
	Source of Failure	−2.798	−7.024 *
	Level of Embarrassment	0.202	2.959 *
	Source of Failure × Level of Embarrassment	0.315	2.931 *

Dependent variables: attribution to the service provider; * represents *p* < 0.05.

**Table 3 behavsci-14-00136-t003:** Confirmatory factor analysis of measurement model.

Constructs/Items	Factor Loading	*t*-Value	Composite Reliability	Average Variance Extracted
Locus Attribution			0.949	0.903
V1	0.939	16.929 *		
V2	0.961	17.555 *		
Level of Embarrassment			0.918	0.789
V4	0.868	15.722 *		
V5	0.892	16.383 *		
V6	0.904	16.754 *		
Negative Emotion			0.723	0.471
V7	0.812	13.36 *		
V8	0.587	8.896 *		
V9	0.639	9.866 *		
Negative Behaviors			0.862	0.676
V10	0.873	15.559 *		
V11	0.781	13.187 *		
V12	0.810	13.925 *		

* represents *p* < 0.05.

**Table 4 behavsci-14-00136-t004:** Results of structural model (H4~H7).

	Path Coefficient	*t*-Value	*R* ^2^
Negative Emotion			0.473
Level of Embarrassment (H4)	0.361	5.459 *	
Locus Attribution (H5)	0.507	7.392 *	
Negative Behaviors			0.720
Locus Attribution (H6)	0.143	2.067 *	
Negative Emotion (H7)	0.792	8.776 *	

* represents *p* < 0.05.

## Data Availability

The raw data supporting the conclusions of this article will be made available by the authors on request.

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
