# Peer review of "Effects of Embarrassment on Self-Serving Bias and Behavioral Response in the Context of Service Failure"

_behavsci, 2024, doi:10.3390/bs14020136_

Round 1

Reviewer 1 Report

Comments and Suggestions for Authors

The paper introduces a rather intriguing topic and focuses on sentiment analysis, namely on embarrassment. The author of the paper has taken a unique and divergent approach from other studies conducted in this field. The literature review presented in the paper is comprehensive and introduces several interesting concepts. These concepts facilitate a better understanding of the topics related to the study. In the meantime, I suggest updating the references. It is not possible to conceptualise such recent subjects with authors from the 70s and 80s. The scenery has changed enormously over the last 40 years.  Also concerning theoretical background, the initial paragraphs of section 2.4 are repetitive and the concepts that have already been introduced in section 2.3. For instance, the FACE Model has been presented earlier and does not require reintroduction.

The paper employs an interesting methodology, which is well-described. The results that have been obtained through this methodology are appropriate for the research purpose. However, there is a need to deepen the discussion presented in Chapter 5. Although there is an explanation of the results, it would be beneficial to understand how these results are associated.

Reviewer 2 Report

Comments and Suggestions for Authors

Thank you for the opportunity to review the manuscript entitled: 'Effects of Embarrassment on Self-Serving Bias and Behavioural Response in the Context of Service Failure.' I enjoyed reading the piece but have the following minor comments:

It would be better to integrate hypotheses with your literature review to demonstrate why you formulated them.

H4 is quite vague to me; could you please revise it?

It is not clear why an external locus of attribution creates negative emotions.

Although the link between negative emotions and behaviour is important to examine, it would be good to consider a moderator that can change the equilibrium, i.e., what can transform negative emotions into positive behaviour.

Additional comments:

How do customers react to different service failures, considering the role of responsibility attribution? The study focuses on the impact of embarrassment and self-serving bias on customers' behavioral intentions.

The topic is relevant. While embarrassment in customer failure situations has been explored before, this study goes beyond by examining scenarios where the service provider takes responsibility for the failure. This novel approach contributes to understanding customer reactions in such situations.

The study's focus on self-serving bias and the attribution of responsibility in service failure cases is a unique approach. It sheds light on why customers may engage in negative behavior and contributes to the existing body of knowledge on this subject.

In the methodology, it is advisable to consider moderators that can influence embarrassment and negative behavior. Exploring how these moderators can transform embarrassment into either positive or neutral behaviors, such as incorporating the Behavioral Approach System (BAS), would enhance the study. 

  Wang, T., Mukhopadhyay, A., & Patrick, V. M. (2017). Getting consumers to recycle NOW! When and why cuteness appeals influence prosocial and sustainable behavior. Journal of Public Policy & Marketing36(2), 269-283.

The conclusions align with the discussed findings and answer the research questions; however, some revisions may be needed in light of suggested changes.

The references are relevant and up to date.

No further comments 

Good luck with your revision.

Reviewer 3 Report

Comments and Suggestions for Authors

The research is very interesting and innovative, and could contribute to the existing knowledge. However, there are some concerns I need to express and the authors should address:

·      The authors have written good introduction to the topic, but some clarifications are still needed. When stating the research gap, they mention that little is known about consumer embarrassment in the service failure context. Introduction should state existing studies (if any) about consumer embarrassment in the service failure context or state that there are no studies concerning this topic. So, the research gap, in this context, should be better elaborated.

·      The introduction, but also the whole paper cites very out-of-date references. Some new, fresh insights should be added. Majority of references are event 30-40 years old.

·      H4. Level of embarrassment positive influences negative emotion should be reformulated to: H4. Higher level of embarrassment positive influences negative emotion.

·      I am concerned if results could be really considered valid as they are based on imaginary and not real time situation. I am also in doubt if students are adequate sample to be chosen for this study, as they are somehow homogeneous group, which is not the case in real time situations.

·      Practical implications should be better elaborated as in this form they are too general.

Comments on the Quality of English Language

The quality of written language is satisfactory and communication is smooth

Round 2

Reviewer 3 Report

Comments and Suggestions for Authors

After checking the revised version of the paper, I consider that the paper is now in good form to be published. The authors have done good work to address all the suggestions and comments.